# Evaluating Dimensional Stability in Modified Wood: An Experimental Comparison of Test Methods

**Rosie Sargent** 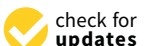

Scion, Titokorangi Drive, Rotorua 3010, New Zealand; rosie.sargent@scionresearch.com

**Abstract:** Dimensional stability is a commonly targeted property for improvement through wood modification. Here four different tests have been performed on three types of modified wood to compare methods of measuring dimensional stability behavior. These tests cover long and short time periods, as well as dimensional changes caused by contact with liquid water, or from changes in air humidity. All the tests showed increased dimensional stability of the modified samples relative to the unmodified controls; however, the relative behavior of the different modifications varied between tests. Soaking in water until maximum swelling showed no differences between thermally modified and furfurylated samples, but a subsequent test showed large differences in the rate of swelling for each wood type, with the furfurylated samples swelling very slowly. Long-term swelling in humid air showed similar results to soaking in water, but with the thermally modified samples having significantly greater dimensional stability than the furfurylated samples. Swelling for a short period in humid air showed no difference in swelling between the modified wood types, but there was a threefold reduction in swelling compared to the unmodified controls. For a more complete understanding of dimensional stability, several tests employing different test conditions should be used.

**Keywords:** acetylation; dimensional stability; equilibrium moisture content; furfurylation; *Pinus radiata*; shrinkage; swelling; thermal modification



## 1. Introduction

In-service changes in wood dimensions can have a significant effect on the performance of wood products, leading to issues such as cracking, distortion, and potentially a loss of function (e.g., jammed doors and windows) [1–3]. Dimensional stability is defined as changes to wood dimensions due to changes in wood moisture content. Changes in moisture content can be caused by different mechanisms—when wood is exposed to liquid water, or water vapor, and the conditions causing the change in moisture content can occur over long or short time periods. All of these will have different effects on the way the wood dimensions respond. Dimensional stability is considered to be one of the most important properties of wood products and is a key property being targeted for improvement in large numbers of research activities worldwide. To enhance the wood dimensional stability, different methods have been developed to modify wood products through changing the way that wood interacts with water. It would not be surprising if different wood modifications perform differently depending on the nature of the change in moisture conditions they face.

The evaluation of wood dimensional stability is of great importance to understand how a modified wood behaves for various applications and under different conditions. There are several methods used for the evaluation of the dimensional stability of modified wood products. Test methods familiar in a scientific context, e.g., the repeated water-soak test to calculate anti-shrink efficiency (ASE), seems harsh and unrealistic to people in industry, because the conditions used (pressure resaturation and oven drying) are not seen in service. On the other hand, the most common metric used by industry "shrinkage from green to 12% MC", is meaningless for modified wood, which is produced from dry

timber, and consequently does not exist in a 'green' state. Indeed, it has been suggested that shrinkage from green to 12% MC is not appropriate for predicting the in-service behavior of any wood [4], because shrinkage behavior over this range is highly non-linear, and the total shrinkage over this range will not accurately predict shrinkage over the smaller changes in moisture content seen in service. The USDA Wood Handbook [5] publishes values of shrinkage from green to oven dry but cautions that these should only be used "if a great deal of accuracy is not required". Figures of shrinkage rates between 6 and 14% MC are also provided, and it is recommended that these are used where possible. In a previously published review on this topic, I explained different methods used for the evaluation of wood dimensional stability [6]. However, there is still a lack of information in the literature about any standardized or widely agreed-on methods for measuring dimensional stability that could be applied to modified wood. It is also important to ensure that dimensional stability tests are capturing the key characteristics that will predict in-service behavior. Most dimensional stability tests measure the maximum changes in dimensions a sample will undergo, but do not measure the rate of dimension change. It has previously been found that thermal modification can decrease the rate of swelling in several European wood species [7], and this is likely to have an impact on how the wood performs in service. With this in mind, it is desirable to examine different test methods that could be applied equally to modified and unmodified wood types, to enable direct comparisons to be performed between new modified wood products and existing wood species that end-users are familiar with.

The current study aims to perform such an experimental comparison to understand the differences between results obtained by measuring wood dimensional stability using a range of test methods.

Four test methods were selected to cover combinations of two different wetting mechanisms (liquid water or water vapor) and two different test durations (short- or long-term exposures). These test methods were as follows:

- Repeated water soaking to saturation followed by oven drying
- Short-term water soaking (swellometer test)
- Long-term (equilibrium) humidity cycling
- Short-term humidity cycling (Harris test),

To compare the results of the different test methods, radiata pine (*Pinus radiata* D.Don) modified via three modification classes has been used to give a range of properties:

- Cell-wall impregnation modification (furfurylation),
- Thermal modification using superheated steam (dry conditions)
- Cell-wall chemical modification (acetylation).

In addition to the modified samples, unmodified control radiata pine samples were tested too. The three modifications were chosen because they each increase dimensional stability via a different mechanism. Furfurylation impregnates furfural alcohol polymer into the cell wall, where it is polymerized to bulk the cell wall and retain it in a swollen state. This increases dimensional stability by preventing the wood from shrinking and reduces hygroscopicity due to the polymer occupying spaces in the cell wall that could otherwise be occupied by water [1]. Thermal modification degrades hemicelluloses, removing OH groups and reducing the hygroscopicity of the wood cell wall [8]. Acetylation substitutes the OH groups in the wood polymers with acetyl groups, which swells the cell wall. This prevents the cell wall from shrinking, and reduces the hygroscopicity, due to the lower OH content [1]. Because of these differences between the modifications, it would not be surprising if each showed different dimensional stability behaviors over the range of tests assessed here. Indeed, some studies have compared dimensional stability behavior of some the three modifications used here and found differences in dimensional stability behavior between the modification types. Dong et al. [9] acetylated and furfurylated wood from four fast-grown species and found that acetylation almost always gave higher ASE values than the same species furfurylated to 70% WPG, but the water uptake of the furfurylated

samples was almost always lower than the acetylated samples. Čermák et al. [10] compared swelling in humid air between thermally modified and acetylated beech wood and found that both modifications significantly reduced the level of swelling compared to unmodified controls, with an ASE value of 49% for thermally modified beech after soaking to maximum moisture content, and 79% ASE for acetylated beech. Swelling anisotropy ratios were calculated for swelling to 90% RH; this showed a significant decrease in anisotropy for the acetylated samples, but not for the thermal modification. When soaked in water for 48 h both the acetylated and thermally modified beech swelled more slowly than unmodified beech and did not swell as much as the unmodified beech. Interestingly the unmodified beech swelled more in the tangential direction compared to the radial direction, but the acetylated beech swelled less in the tangential direction.

In this work the implemented conditions have been described for each test method, and the obtained results from the different tests, and the different wood modifications have been compared to each other.

## 2. Materials and Methods

### 2.1. Wood Modifications

Four green radiata pine (control) boards (100 × 40 mm cross-section) were obtained from a sawmill in the central North Island of New Zealand and kiln dried using a typical appearance-grade schedule in a lab-scale kiln.

Furfurylated radiata pine ('FA') was produced at Scion using four matched boards of radiata pine and a furfuryl alcohol formulation developed in-house at Scion. The boards (100 × 40 mm cross-section) were impregnated with a furfuryl alcohol solution containing 5% dilactide catalyst using a Lowry empty-cell impregnation process, then cured to a final weight gain of around 60%. Furfuryl alcohol and dilactide were sourced from Sigma-Aldrich, Auckland, New Zealand.

ThermoWood ('TH') thermally modified radiata pine [11] was purchased from Tunnicliffe Timber in Edgecumbe, New Zealand. Two boards were supplied with a 100 × 40 mm cross-section. The exact modification schedule is not known, but it is likely to have been modified to around 230 °C.

Accoya ('AC') acetylated radiata pine [12] was produced by Accsys in Arnhem, The Netherlands and purchased from ITI Timspec in Auckland, New Zealand. Two boards were supplied with a 200 × 50 mm cross-section. As this is a commercial product, the final weight gain from the acetylation is not known but is expected to be 21–23%.

### 2.2. Repeated Water-Soak Test

A repeated water-soak test was performed to calculate the ASE values of the samples. Two 15 mm 'biscuits' were cut from near one end of each board to be tested (4 boards each for control and FA, 2 boards each for AC and TH). The biscuits were cut to the full cross-sectional dimension of each wood type (100 × 40 mm for control, FA and TH and 200 × 50 mm for AC). While it is not generally good practice to mix specimen dimensions within one test, it was decided to maximize the dimensions of each specimen, to minimize the effect of measurement errors, especially for the Accoya samples, which were expected to swell by a very small amount. Choosing specimen dimensions for dimensional stability tests is often a tradeoff between maximizing dimensions to minimize measurement errors and minimizing dimensions to improve sample uniformity (e.g., removing the effect of growth-ring curvature). The biscuits were resaturated using a vacuum–pressure–soak method [1]. For that, the biscuits were submerged in water, then vacuum (−85 kPa) was applied for 15 min. Then, pressure was applied to 175 kPa for 1 h, followed by an atmospheric pressure soaking for 48 h. After soaking, the radial, tangential, and longitudinal dimensions of the biscuits were measured with digital calipers (Mitutoyo Absolute Digimatic Caliper, Aurora, IL, USA). This impregnation method is very similar to a 'tank' vacuum/pressure impregnation step often used in small scale wood modification or preservative treatment.

The biscuits were then stacked on oven racks in a laboratory kiln and dried overnight using a 50/40 °C (dry bulb/wet bulb) schedule and a low air flow. The oven racks were then transferred to a laboratory oven and the biscuits were dried at 103 °C until constant weight. The biscuit dimensions were measured again. This drying regime was chosen to minimize checking (cracking) of the samples, which can occur during this test, and which can compromise the test results [13].

Two further water-saturation/oven-dry cycles were completed with these biscuits. *ASE* is calculated according to Equation (1).

$$ASE = \frac{(V_{UWS} - V_{UOD})/V_{UOD} - (V_{TWS} - V_{TOD})/V_{TOD}}{(V_{UWS} - V_{UOD})/V_{UOD}} \times 100 \qquad (1)$$

where:

$V_{UWS}$—volume of unmodified reference biscuit after water saturation; [mm$^3$]
$V_{UOD}$—volume of unmodified reference biscuit after oven drying; [mm$^3$]
$V_{TWS}$—volume of modified biscuit after water saturation; [mm$^3$]
$V_{TOD}$—volume of modified biscuit after oven drying; [mm$^3$]

Normally the $(V_{UWS} - V_{UOD})/V_{UOD}$ term would be determined from unmodified 'reference' samples cut from the same material as the modified wood. This gives more accurate results and it is relatively straightforward to source suitable unmodified material when modifying wood samples in the lab. Because the wood used here came from a range of sources, using matched reference material was not possible. In order to illustrate the kinds of results that are obtained when unmodified reference samples are used, an appropriate reference swelling value was required. The average swelling of the control samples could have been used, but for consistency across all the modifications, it was decided to use a published figure of 11.6%. This was calculated from published individual tangential, radial and longitudinal shrinkage values [14], which were converted to percentage swelling values (percentage swelling is calculated with respect to the oven dry dimensions, whereas percentage shrinkage is calculated with respect to the water saturated dimensions). It should be noted that the published shrinkage values were measured from a green (never dried) condition, which may have larger dimensions than the resaturated conditions being measured here. Where matched unmodified reference samples are available, it is recommended these be used for calculating the percentage swelling. Where these are not available, it is recommended that an alternative metric, such as the percentage volumetric swelling, or percentage tangential be calculated instead. While the percentage volumetric swelling (%S) is widely reported [5,13,15], often dimensional stability issues are caused by movement in one particular grain direction, rather than the total change in wood volume, so a direction based measurement such as the percentage radial or tangential swelling may give more useful information about the wood behavior.

The method development for this test has been reported previously [16]. From this work and others (e.g., [13,17]), it was found that the first water soak/oven-dry cycle often gives different results to subsequent cycles, so here the ASE values from the first cycle have not been included in the subsequent analysis.

The swelling anisotropy ratio (ratio of tangential to radial dimension change) can be calculated from the individual measurements that make up the volume terms in Equation (1), and is defined as follows:

$$T/R = \frac{(T_{ws} - T_{OD})/T_{OD}}{(R_{ws} - R_{OD})/R_{OD}} \qquad (2)$$

where:

$T/R$—swelling anisotropy ratio
$T_{WS}$—water saturated tangential dimensions; [mm]
$T_{OD}$—oven dried tangential dimensions; [mm]
$R_{WS}$—water saturated radial dimensions; [mm]
$R_{OD}$—oven dried radial dimensions; [mm]

### 2.3. Short-Term Water Soak (Swellometer) Test

The swellometer test is based on the method specified by the US Window and Door Manufacturers Association (WDMA, Washington, DC, USA) [18]. This standard specifies samples of 127 mm or 254 mm in the tangential direction. But wood samples of this dimension were not available for the present study. Therefore, two samples of 38 × 100 × 6 mm (radial × tangential × longitudinal dimensions) were cut from each board to be tested, and equilibrated at 25 °C, 65% relative humidity (RH) for 5 weeks.

Samples were then loaded into a swellometer jig (Figure 1), which consists of a rigid back, which supports a digital dial gauge (Mitutoyo Aboslute Digimatic Indicator, Aurora, IL, USA), and a channel that the wood slides into. The wood is fixed against the end of the dial gauge by a pair of brass stops that slide into the channel and can be fixed in place via a screw. One side of the channel can be adjusted sideways to accommodate different widths of samples. The channel restrains the sample sufficiently, so it remains in the correct orientation during the test but leaves enough space for the sample to swell without its movement being restricted by the channel.

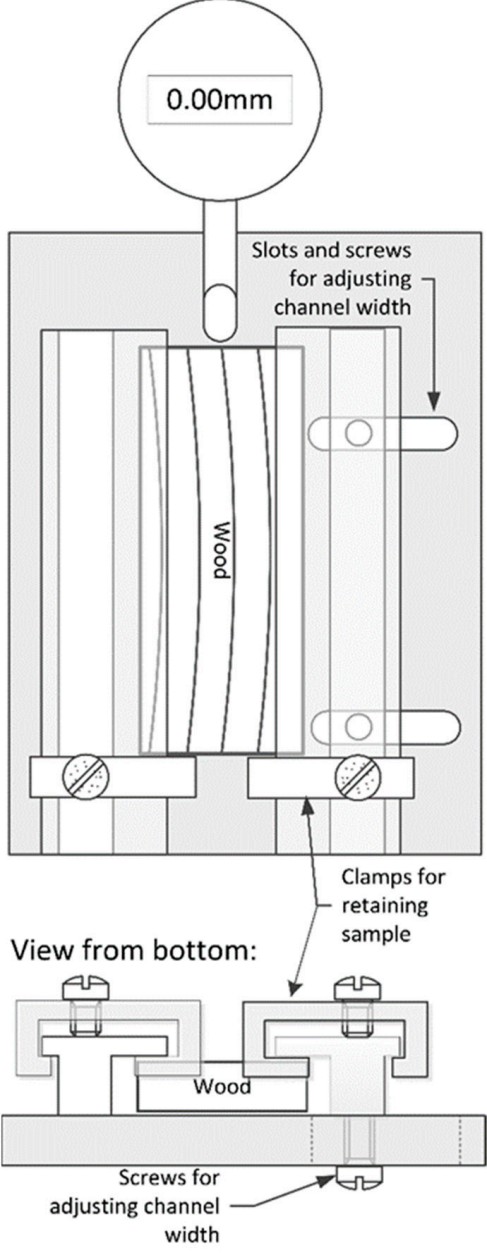

**Figure 1.** Front and bottom views of swellometer jig.

For the standard test method, the initial tangential dimension is recorded, then the jig is immersed in distilled water at 24 ± 3 °C and after 30 min the test is stopped, and the length of the tangential dimension is recorded again. It was found that 30 min was not enough time for significant swelling to occur in some samples. So, in this study, the tangential dimension was measured every 5 s during immersion and the test continued for three days, by which time all the samples had stopped swelling.

From the recorded data, the displacement at 30 min can easily be extracted for use in the standard WDMA calculations. According to the standard, the anti-swelling effectiveness ($ASE_W$) is calculated for each sample as follows:

$$ASE_W = \frac{(T_{U30} - T_{U0}) - (T_{M30} - T_{M0})}{(T_{U30} - T_{U0})} \times 100 \tag{3}$$

where:

$T_0$—tangential dimension before soaking; (mm)

$T_{30}$—tangential dimension after 30 min soaking; (mm)

Additional subscripts M and U indicate modified and unmodified reference samples, respectively. This metric is also sometimes referred to as the water repellent effectiveness (WRE) [19].

As with the repeated water-soak test, unmodified reference material that was matched across all wood types was not available here. Normally, a comparative metric such as ASEw would only be used where suitable reference measurements were available, but here it has been included for illustrative purposes. The average swelling of the control samples could have been used as the $(T_{U30} - T_{U0})$ term in Equation (3), but to use a consistent basis for comparison across all the modifications, it was decided to use a published value of tangential swelling of radiata pine between 12% MC and green (4.2%) [14]. As with the repeated water-soak test, wood in the green state is not directly equivalent to resaturated wood, so the level of swelling seen here may be slightly lower than the values calculated from green samples. Normally a comparative metric such as $ASE_w$ would only be used where a relevant unmodified reference was available (e.g., matched modified and unmodified samples). In this study matched reference samples were not available for all the treatments, so a published value was used for consistency.

In addition to calculating $ASE_w$, two additional metrics were chosen: the percentage of maximum swelling after 3 days ($\%SW_{max}$), and the percentage of swelling that occurs after 30 min ($\%SW_{30}$). These are intended to quantify the overall levels of swelling of the samples, and the rate at which they swell, respectively.

Maximum swelling is defined as the percentage difference between the initial sample dimension and the final sample dimension. As the tests were run until all samples had finished swelling, this final dimension is the maximum sample dimension.

$$\%SW_{max} = \frac{\left(T_{final} - T_0\right)}{T_0} \times 100 \tag{4}$$

where:

$SW_{max}$ is the maximum swelling (% of initial tangential dimension)

$T_{final}$ and $T_0$ are the final and initial tangential dimensions (mm), respectively.

$\%SW_{30}$ is the proportion of the swelling that occurs after 30 min soaking, and is calculated as follows:

$$\%SW_{30} = \frac{T_{30} - T_0}{T_{final} - T_0} \times 100 \tag{5}$$

where:

$T_{30}$ is the tangential dimension (mm) after 30 min soaking, and the remaining parameters are defined in Equations (3) and (4).

*2.4. Equilibrium Humidity Cycling Test*

This test was performed based on the European standard DIN 52 184 [20]. Two samples of 30 × 30 × 10 mm (radial × tangential × longitudinal) were cut from each of the boards to be tested. The DIN standard involves taking one set of dimension measurements each at two different humidity levels. Here, the aim was to look at the effect of cycling humidity over time. So, this study took the standard DIN method and extended it to perform many more measurements at multiple humidity levels.

The blocks were placed in a controlled environment at 25 °C, 65% RH, until constant mass was attained (defined as less than 0.1% change in mass over 24 h). The DIN standard specifies 20 °C, 65% RH, but the controlled humidity rooms used at Scion are all maintained at 25 °C. So, 25 °C was used for all the humidity test conditions.

The radial and tangential dimensions of the blocks were measured using a digital dial gauge (accurate to 0.001 mm, Sylvac Digital Indicator, Yverdon, Switzerland), which was firmly mounted on a bench to prevent movement during measurement. The block sat flat against the base of the measurement jig and was held firmly against two measurement pins opposite the dial gauge. The block could then be moved sideways until the dial gauge aligned with a line marked in felt-tipped pen 10 mm from one corner of the block. This method enabled accurate and repeatable measurement of the same locations on each block for every measurement.

After the initial dimension measurement, the samples were placed in the first humidity environment listed in Table 1, until their weight had stabilized. Then, dimensions were measured again according to the method above. The samples were then conditioned and measured at each of the three remaining environments listed in Table 1. These four steps constitute one full humidity cycle. Two further humidity cycles were completed. Then, the blocks were oven dried at 103 °C to constant weight and the weight and dimensions recorded again. Because the wood behavior during the first humidity cycle is often different to all the subsequent cycles [6], only data from the second and third humidity cycles were included in the subsequent analysis.

**Table 1.** Conditions used for equilibrium humidity cycling.

| Step | Temperature | Relative Humidity |
| --- | --- | --- |
| 1 | 25 °C | 60–70% RH (High) |
| 2 | 25 °C | 90–95% RH (High) |
| 3 | 25 °C | 60–70% RH (Low) |
| 4 | 25 °C | 60–70% RH (Medium) |

From these data, the following calculations can be made for each humidity step:
Equilibrium moisture content (*EMC*)

$$EMC = \frac{m_{MC} - m_{OD}}{m_{OD}} \times 100 \qquad (6)$$

Tangential dimensional change (swelling)

$$\Delta T = \frac{T_{MC} - T_{OD}}{T_{OD}} \times 100 \qquad (7)$$

where:
   *EMC*—equilibrium moisture content; (%)
   $m_{MC}$—mass at specified humidity; (g)
   $m_{OD}$—mass when oven dry; (g)
   $\Delta T$—change in tangential dimension from oven dry; (%)
   $T_{MC}$—tangential dimension at the specified humidity; (mm)
   $T_{OD}$—tangential dimension when oven dry; (mm)
   Radial dimensional change ($\Delta R$) is calculated in the same way.

These measurements are relative to oven-dry dimensions, which do not have a lot of relevance to an in-service situation, where the wood will alternate between periods of high and low air humidity, without ever being oven dried. As a more useful measure of the change in dimension, the swelling coefficient 'h' can be calculated. This is defined as the change in dimension for each 1% change in relative humidity. Skaar called this the humidity expansion coefficient [2] and suggested that this metric is much more important from a wood utilization standpoint than the change in moisture content for a given change in humidity. In DIN 52 184, the swelling coefficient is calculated from two measurements made at different humidity levels, as follows:

$$h_T = \frac{T_{MC1} - T_{MC2}}{T_{OD}(RH_{MC1} - RH_{MC2})} \times 100 \tag{8}$$

where:

$h_T$—swelling coefficient in the tangential dimension
$RH_{MC1}$—relative humidity at first measurement level; (%)
$RH_{MC2}$—relative humidity at second measurement level; (%)
And all other terms are as defined above.

Here multiple measurements were taken at different humidity levels. So, the swelling coefficient has been calculated as the slope of a line fitted between the dimensional change values calculated in Equation (2) and the relative humidity level at which they were measured. Both adsorption and desorption data were fitted at once, making the swelling coefficient an average of the adsorption and desorption behavior. An example of this analysis is included in the supplementary data file.

The swelling anisotropy ratio can be calculated from the radial and tangential dimension changes as follows:

$$T/R = \frac{(T_{MC} - T_{OD})/T_{OD}}{(R_{MC} - R_{OD})/R_{OD}} \tag{9}$$

where:

$T/R$—swelling anisotropy ratio
$T_{MC}$—equilibrated tangential dimensions; (mm)
$T_{OD}$—oven dried tangential dimensions; (mm)
$R_{MC}$—equilibrated radial dimensions; (mm)
$R_{OD}$—oven dried radial dimensions; (mm)
In this case, where multiple measurements have been made at different humidity levels, the swelling anisotropy ratio is equivalent to the slope of tangential vs. radial swelling. So, the swelling anisotropy ratio for each specimen can be calculated using linear regression.

### 2.5. Short-Term (Harris) Humidity Cycling Test

This test is based on the method of J.M Harris [21]. Unlike the previous tests, this test does not aim to measure wood swelling under equilibrium conditions but quantifies how much a specimen will swell in a given time frame. The original test used samples 4″ × 1$\frac{1}{4}$″ × 5/16″ (approx. 102 × 32 × 8 mm), to give a similar ratio of tangential and radial surfaces to a standard 4″ × 2″ board. For simplicity, this study used specimen dimensions that match the equilibrium humidity cycling test described above. The identical dimensions in the radial and tangential directions give the advantage of having similar levels of measurement error in each direction. Two specimens were cut from each board to be tested and were prepared as for the equilibrium humidity cycling test, including the markings for dimension measurements. After equilibrating in a medium humidity environment (25 °C, 65% RH), the faces (i.e., end grain) of the blocks were painted with a two-pot epoxy paint (Carboguard 635, Altex Coatings, Tauranga, New Zealand) to prevent moisture movement through the faces. The edges were protected with adhesive tape during painting to keep them free of paint. After painting, the blocks were equilibrated at 25 °C, 65% RH a second

time and their weights and dimensions were recorded according to the method described in the equilibrium humidity test.

The blocks were then placed in a high humidity environment (25 °C, 80–90% RH) for 24 h and their weights and dimensions were measured again. The blocks were then returned to the medium humidity environment until their weights stabilized and were weighed and their dimensions were measured. This completed one humidity cycle. Two further humidity cycles were performed. Then, the blocks were oven dried at 103 °C until constant weight and their weight and dimensions were recorded.

The tangential dimension change can be calculated as follows:

$$\Delta T = \frac{T_{90} - T_{65}}{T_{65}} \times 100 \tag{10}$$

where:

$\Delta T$—change in tangential dimension; (%)

$T_{90}$—tangential dimension at high humidity; (mm)

$T_{65}$—tangential dimension at medium humidity; (mm)

Radial dimensional change ($\Delta R$) is calculated in the same way.

For consistency, the results from the first humidity cycle were discarded, as with the equilibrium humidity cycling test, and the repeated water-soak test described before.

### 2.6. Statistical Analysis

All statistical analysis was performed using the RStudio software (Version 4.1.1 RStudio Team. Boston, MA, USA.) [22].

Analysis of variance (ANOVA) was used to compare the mean values of each dimensional stability metric, and differences between wood types were calculated using Tukey's honest significant difference (HSD) test using a 95% confidence level.

Where several dimensional stability metrics could be calculated from the same data, linear regression was used to model the relationship between these variables to determine if they would give equivalent results.

For the equilibrium humidity cycling test, swelling coefficients were calculated for each individual sample using linear regression to model the slope of dimensional change ($\Delta T$ or $\Delta R$) vs. RH. Any samples with a poor model correlation ($p > 0.05$) were excluded from the subsequent analyses. Anisotropy ratios were calculated in the same way, modelling $\Delta T$ as a function of $\Delta R$.

For the swelling anisotropy ratios calculated in the repeated water-soak test, each wood type was compared to a published swelling anisotropy ratio [14] using a *t*-test for the ratio of two means [23,24], with the null hypothesis that the calculated ratio was equal to the published figure.

## 3. Results

### 3.1. ASE from Repeated Water-Soak Test

Anti-shrink efficiency (ASE) results are shown in Figure 2. All the modifications had significantly higher ASE values than the control samples, indicating improved dimensional stability. The Accoya samples (AC) had the highest ASE values, indicating a substantial reduction in dimension changes between the water-saturated and oven-dried states. The furfuryl alcohol formulation samples (FA) and ThermoWood samples (TH) had lower ASE values, but these were still significantly higher than the control samples. As noted in the methods section, ASE would normally be calculated relative to unmodified wood from the same source as the modified wood. Often, it is not possible to have such a single 'reference' wood type that is relevant to all the treatments being compared (such as in this study where commercially modified wood is being tested). In these cases, it makes more sense to use a metric that does not require unmodified reference values. For example, the percentage of dimensional changes (radial, tangential, or volumetric) of each wood type can be compared directly. For the data presented here, there is a strong correlation between the percentage

tangential swelling and ASE ($R^2$ = 0.922), so the percentage tangential swelling would be a more suitable metric to use in situations where calculating ASE by comparing all the samples with an unmodified reference does not make sense (e.g., for commercial samples, or comparing wood species). Further details of this regression analysis can be found in the supplementary data file.

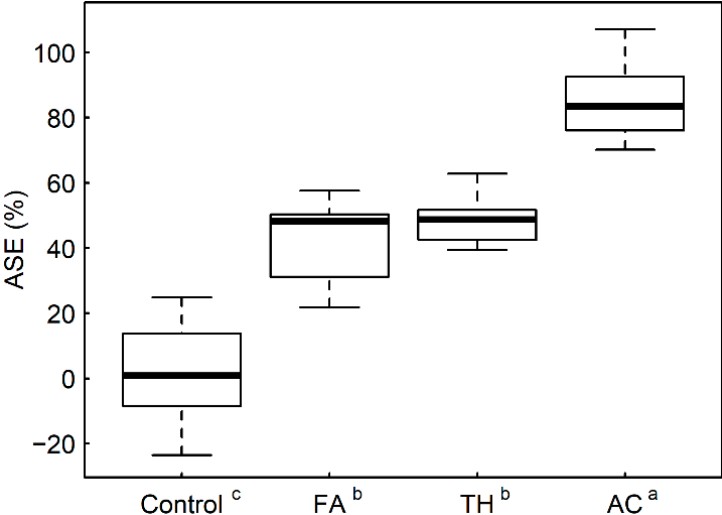

**Figure 2.** Anti-shrink efficiency values for each modification type (excluding the first water-soak oven-dry cycle). Superscripted letters on the *x*-axis indicate significant differences according to Tukey's HSD test (95% confidence level).

The results obtained for the swelling anisotropy ratios (between oven dry and water saturated dimensions) are shown in Table 2. Because the AC samples swelled very little (as seen by the high ASE values in Figure 2) the dimensional changes being measured were a similar magnitude to the random errors in the measurements, leading to a high degree of scatter in the data—particularly for the radial dimensions, which swelled by an average of 0.15 mm in the AC samples, compared to an average of 1.4 mm swelling for the control samples. These very small and variable swelling values in the AC samples led to extremely variable swelling anisotropy ratios (as seen by the high standard deviation value in Table 2), making the ratio meaningless for this wood type. Interestingly, the control and TH samples have a significantly lower swelling anisotropy ratio to the published value of 2.1 [14]. Additionally, the swelling anisotropy ratio for the TH samples is significantly lower than the control samples ($p < 0.01$). The FA samples also have an average ratio below 2.1 but this difference is not statistically significant.

**Table 2.** Mean swelling anisotropy ratios (T/R) for each wood type, plus standard deviation values (SD) in parentheses.

| Wood Type | T/R Ratio (SD) |
|---|---|
| Control | 1.91 (0.28) * |
| FA | 1.91 (0.46) |
| TH | 1.54 (0.26) * |
| AC | 3.97 (4.04) |

* indicates ratios that are significantly different to the published value of 2.1 (95% confidence level).

### 3.2. Swellometer Test

The swellometer anti-swelling effectiveness ($ASE_w$) values for each wood type are shown in Figure 3. As with ASE above, $ASE_w$ measures the percentage reduction in swelling relative to unmodified wood, so higher values indicate greater dimensional stability. Ideally the average swelling of the control samples would be the same as the published swelling

figure, and thus the average $ASE_w$ of the control samples would be zero. Here the control samples have swelled more than the published figure, so almost all control samples have a negative $ASE_w$. This illustrates the importance of having an appropriate unmodified reference when calculating $ASE_w$, and why it is worth considering an alternative metric when an appropriate reference is not available. All the wood types have significantly higher $ASE_w$ values compared to the control samples. The TH samples have significantly lower $ASE_w$ values compared to the other modifications.

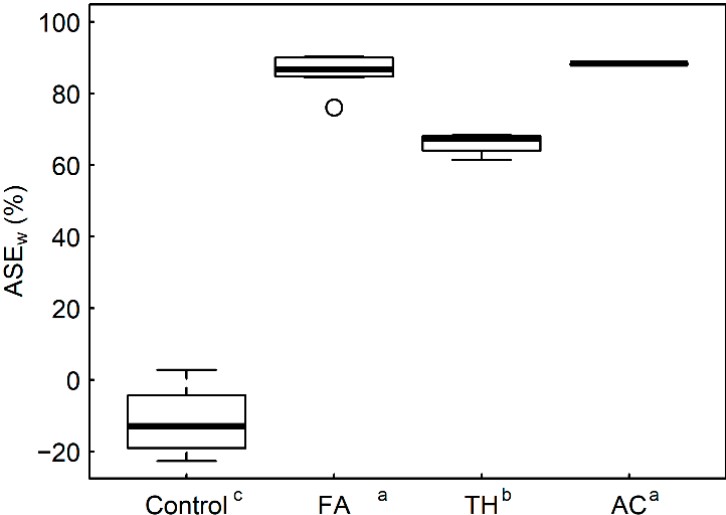

**Figure 3.** $ASE_w$ values for each wood type (calculated following 30 min soaking). Superscripted letters on the *x*-axis indicate significant differences according to Tukey's HSD test (95% confidence level).

Some aspects of these results are quite different to both the ASE results above, and the equilibrium humidity cycling tests (in the following section), where the FA samples were significantly less stable than the AC samples. This suggests that for some wood types, relative dimensional changes can be very different over short time periods compared to longer (equilibrium) time periods. Longer term behavior has previously been defined as 'dimensional stability' and the short-term behavior as 'water repellency' [19], although this terminology does not appear to be in wide use.

After 3 days of water soaking, all the samples had attained their maximum water-soaked dimensions. There was a good correlation between the percentage swelling to these maximum dimensions and the maximum tangential swelling of the repeated water-soak test samples ($R^2 = 0.850$), showing that the swellometer samples are at, or close to, full saturation at the end of the test. As with the water soak/oven dry test, comparing the maximum tangential swelling has the advantage of not requiring unmodified reference measurements.

To further understand the rate of swelling of the different modifications, the proportion of the final swelling that had occurred in the first 30 min soaking ($\%SW_{30}$) is shown in Figure 4. The unmodified control samples had swelled to over 90% of their final dimensions in the first 30 min, indicating that they swelled very quickly. The FA samples, on average, only swelled to around 40% of their final dimension, showing that they swelled much more slowly. The AC and TH samples swelled to 70–80% of their final dimensions after 30 min of soaking. It is interesting to note the relative performance of the FA and TH modifications between the ASE results (Figure 2) where both wood types have similar results (indicating similar levels of swelling), and results from swelling after 30 min (Figure 4), where the FA samples have swelled much less than the TH. If soaked for long enough, both wood types will swell a similar amount, but they do so at very different rates. Examples of tangential swelling over time for each wood type are given in the supplementary data file.

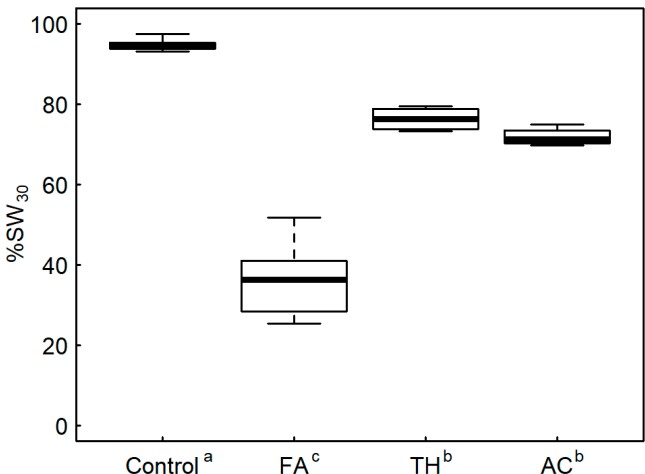

**Figure 4.** Percentage of total swelling that has occurred after 30 min (%SW$_{30}$). Superscripted letters on the *x*-axis indicate significant differences according to Tukey's HSD test (95% confidence level).

*3.3. Equilibrium Humidity Cycling*

The equilibrium moisture content (EMC) at each humidity condition is shown in Figure 5. While EMC is not a measure of stability itself, reduction in EMC following wood modification can be a broad indicator of improved dimensional stability, because it shows that the relationship between the water and the wood has been altered. Reduced moisture content can also have other benefits, such as reduced susceptibility to fungal decay [25]. All modifications have lower average EMC values compared to the control samples. While many measures of dimensional stability equilibrate samples at standard conditions (20 °C 65% RH) as a reference point, some measures of dimensional stability are based on dimensions at 12% moisture content or describe dimensional changes in terms of a certain change in wood moisture content. As can be seen from Figure 5, the TH and FA samples would only reach 12% moisture content at a very high relative humidity, and the AC samples would possibly be above fiber saturation at 12% MC—both of these are very different moisture states to that of unmodified wood at 12% EMC. This highlights the difficulty in using moisture content-based metrics for comparing dimensional stability in modified wood products. It should also be noted that chemical and impregnation modifications do add considerable mass to the samples, and this will reduce the apparent moisture content of the wood, due to the increased oven dry mass.

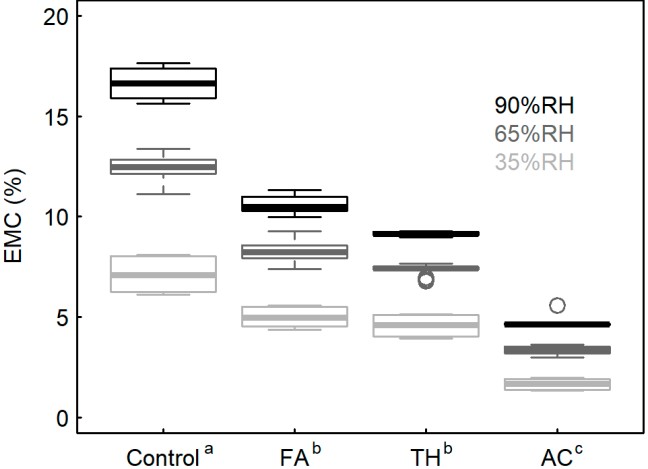

**Figure 5.** Equilibrium moisture content (EMC) values measured for each sample, and at different humidity levels (excluding the first humidity cycle). Superscripted letters on the *x*-axis indicate significant differences according to Tukey's HSD test (95% confidence level). EMC values for all humidity conditions were pooled when calculating the significance groupings.

The results obtained for changes in dimensions (radial and tangential) for a 1% change in relative humidity are given in Figure 6. These results show a similar trend to the EMC values presented above, with the control samples showing the most movement with changes in relative humidity and the AC samples showing the least. TH and FA samples are in the middle. The trend is the same for both the radial and tangential directions, with the absolute values are much smaller in the radial direction, as would be expected. In the tangential direction, the FA modification has a significantly higher average value of swelling coefficient than the TH. But in the radial direction, the difference is not significant for these two samples. This contrasts with the ASE results, where both wood types had a similar average ASE.

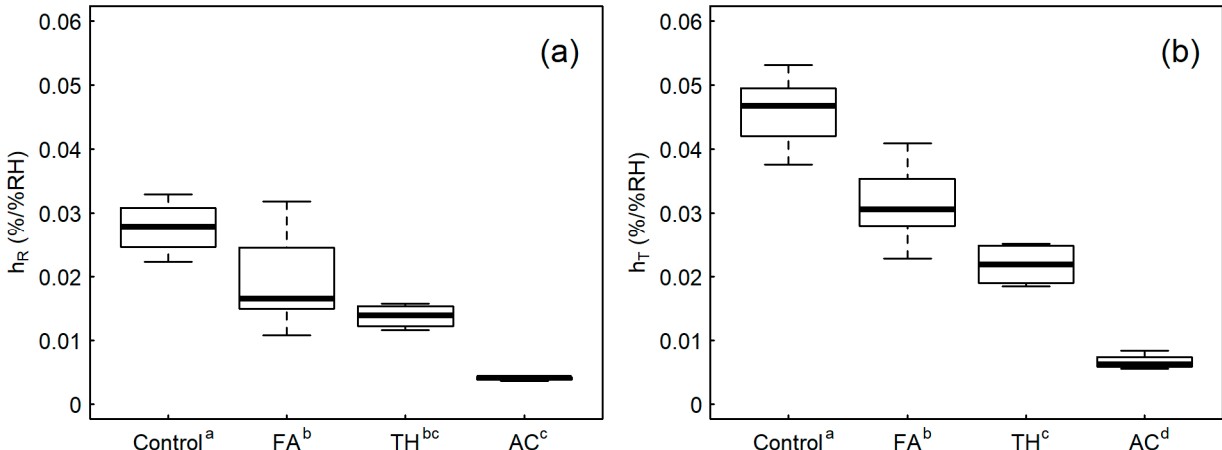

**Figure 6.** Radial (**a**) and tangential (**b**) dimension change for each 1% change in relative humidity ($h_R$ and $h_T$, respectively). Superscripted letters on the *x*-axis indicate significant differences according to Tukey's HSD test (95% confidence level).

Swelling anisotropy ratios can be calculated from the dimensional changes between 90% RH and oven-dryness. This method suffers similar issues to that of the repeated water-soak test described above, so an alternative method of calculating the swelling anisotropy ratio is to calculate the ratios of the tangential and radial swelling coefficients for each sample [4,20]. For the data presented here, this is equivalent to the slope of a line fitted to all the radial and tangential swelling values measured on each sample. Compared to measuring the ratio of only two pairs of measurements, this includes several measurements on the same specimen and is thus less sensitive to random errors in any single measurement. Linear regression was used to calculate the slope of the line for each sample and specimens were excluded from subsequent analysis if the regression relationship between radial and tangential swelling was not statistically significant (95% confidence level). This resulted in two out of four AC specimens being excluded from the analysis, because for these samples, the level of scatter in individual radial and tangential swelling values was so high that no significant correlation between radial and tangential swelling could be found. Examples of this data analysis, including both significant and nonsignificant regression relationships are shown in the supplementary data file. The swelling anisotropy ratios calculated by this method are shown in Figure 7. No significant differences in mean anisotropy ratio were seen between the different wood types (95% confidence level). In contrast with the repeated water-soak test, this method did produce swelling anisotropy values for AC that had a similar level of variability to the other wood types (compared with the high standard deviation seen in the AC values in Table 2). This suggests that the ratio of swelling coefficients is a more robust method to use for wood types with very high dimensional stability. The swelling anisotropy ratio is not constant over the entire humidity range; namely, at very high relative humidity levels (>90% RH) tangential swelling increases at a much greater rate than radial swelling, increasing the swelling anisotropy ratio [26]. For this reason, the swelling anisotropy ratios calculated here should be lower than for

swelling from oven dry to green. Thus, it is not surprising that the anisotropy ratios calculated here are lower than the values calculated from the repeated water-soak test and are lower than the published value of 2.1 for swelling from oven dry to green given in the literature [14]. Previously published work comparing thermally modified and acetylated European beech [10] measured T/R ratios (expressed as a radial to tangential ratio in their study) at 90% RH and saw a significant decrease in T/R ratio for the acetylated beech (T/R of 1.56) compared to unmodified beech (T/R of 1.89).

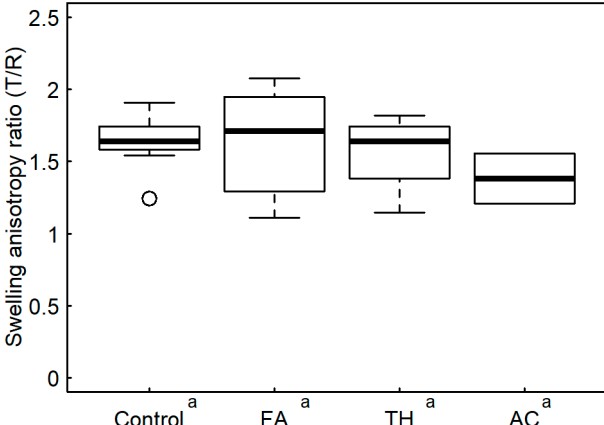

**Figure 7.** Swelling anisotropy ratios (T/R) calculated from the ratio of swelling coefficients. Superscripted letters on the *x*-axis indicate significant differences according to Tukey's HSD test.

### 3.4. Harris Humidity Cycling

Changes in the radial and tangential dimensions after 24 h at 90% RH are shown in Figure 8. In both directions, the control samples show the greatest dimension change, and this is significant compared to all the modifications, showing all the modifications significantly decrease the rate of swelling relative to the control samples. This is especially evident in the tangential direction where the average swelling in the control samples was over three times that in the modified samples.

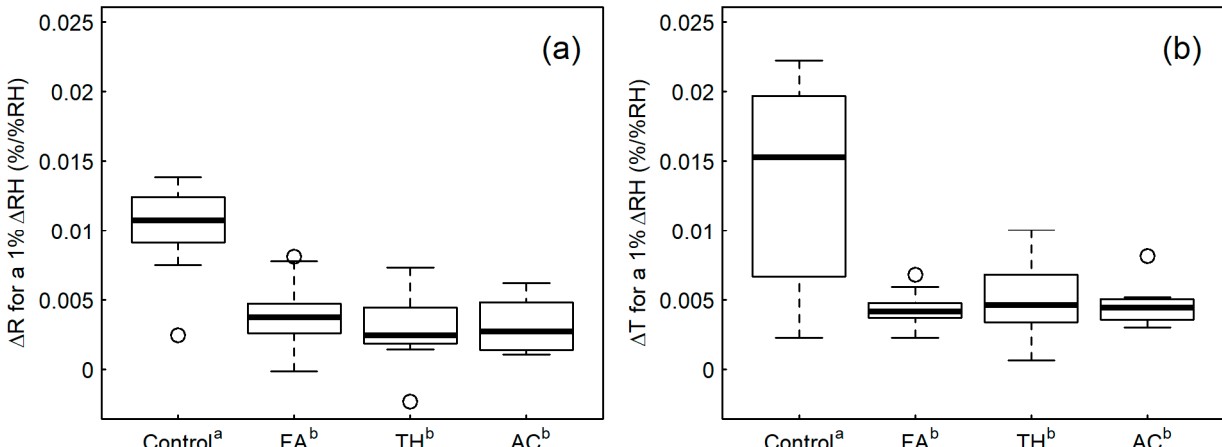

**Figure 8.** Percentage change in (**a**) radial and (**b**) tangential direction after 24 h at 90% RH for a 1% change in RH. Superscripted letters on the *x*-axis indicate significant differences according to Tukey's HSD test (95% confidence level).

Levels of swelling for the three modifications are not significantly different to each other, which contrasts with the swelling coefficient values in Figure 4, where there were significant differences in dimensional stability between the three modifications. This may be due to the relatively high degree of experimental error in the Harris test, or reflect differences in short term swelling behavior in humid air compared to liquid water.

## 4. Discussion

### 4.1. ASE Results Obtained from Repeated Water-Soak Test

The three types of modified wood all had significantly lower ASE values than the unmodified control samples, indicating improved dimensional stability. Additionally, the test showed significant differences in dimensional stability between the AC samples and the other modifications.

One disadvantage of calculating ASE is the requirement for all samples to be compared to the same unmodified reference. For lab-based wood modifications, all the samples are typically from the same batch of timber, so obtaining matched reference samples is very straightforward, and this gives a robust comparison between the dimensional stability before and after modification. When doing comparisons that include commercially modified wood, or when making comparisons between different wood species, identifying a suitable unmodified reference that can be compared to all the samples being tested can be problematic, and may not make practical sense (here a published swelling figure was used, but this is not recommended in practice). A good correlation was seen between the ASE values and the percentage tangential swelling, suggesting that the percentage tangential swelling would be a suitable substitute for situations where a suitable reference material cannot be found, and ASE values cannot be calculated. One example of this is when comparing dimensional stability of different wood species. The percentage tangential swelling has the additional advantage of directly describing the dimensional changes of the wood, giving an indication of how the wood will perform in service. However, it is worth noting that the conditions used in this test (pressure resaturation and oven dryness) will not be seen in service.

Calculation of swelling anisotropy ratios was problematic, especially for AC samples, which showed very small dimensional changes. Two of the wood types had swelling anisotropy values, which were significantly different to published figures for radiata pine, including, interestingly, the unmodified control samples. The difference in anisotropy was not large (mean of 1.9 compared to published figure of 2.1), but this was statistically significant. The mean swelling anisotropy ratio of the AC samples was very high, and the standard error was even higher, calling into question the reliability of these results. Calculating anisotropy ratios from measurements under only two different conditions makes the calculated ratios sensitive to measurement errors, which can lead to such large variability that it obscures any underlying trends. For this reason, the calculation of swelling anisotropy ratios from the repeated water-soak test is not recommended.

The repeated water-soak test has the advantage of being a relatively quick test that does not require a lot of specialist equipment. From the results shown here it also shows a broad indication of the effectiveness of a wood modification to improve dimensional stability.

### 4.2. Swellometer Test

The swellometer test presented here has two key refinements from the WDMA standard: the test is run for a much longer time period (3 days, compared to 30 min in the standard) and displacement is measured continuously. These allow for a more complete understanding of how quickly, and by how much, wood will swell when it comes in contact with liquid water. Comparing the relative performance of the different wood types, the results seen in this test are different to those seen in the repeated water-soak test. After 30 min of soaking, all the modified wood types swelled significantly less than the control samples, but the level of swelling in the FA samples was very similar to the AC samples, and significantly lower than the TH samples. The rate of swelling was also reduced in the modified samples compared to the control samples, but this time the FA samples had the lowest average swelling and the TH and AC samples had higher swelling rates that were similar to each other. By comparing Figures 2 and 4, it can be seen that given enough time, the FA samples will swell almost as much as the TH samples, but for short periods of wetting followed by drying the FA samples will change dimension less than the TH samples. Often in service, wood will be exposed to liquid water for periods of less than

three days (e.g., a day or two of rainy weather), so differences in the rate at which boards swell over these time periods would potentially lead to differences in apparent dimensional stability. This will also potentially reduce the period of time in which the wood is at a high enough moisture content to sustain fungal decay [19]. Expressing results in terms of $ASE_w$ has similar limitations to the ASE results above, namely the requirement for an unmodified reference sample as well as the results being dimensionless, and consequently difficult to relate to in-service performance. The total swelling, and percentage swelling after 30 min, appear to give more meaningful results. However, it should be noted that swelling rates will vary with specimen geometry, because water is preferentially absorbed through the end grain of the samples [7], so measured swelling rates may not accurately predict the rate of swelling of a particular component in service.

The swellometer test has a slower turnaround time than the repeated water-soak test, due to the requirement for samples to be equilibrated prior to testing, and the need for specialist measurement equipment, limiting the number of samples that could be tested in parallel. For the samples tested here, testing took a total of 8 weeks to complete, compared to 2 weeks for the repeated water-soak tests. Despite this, once the swellometer testing is underway it is not labor intensive, and the sample analysis is relatively straightforward. Because there is no need for oven drying, cracking of the test specimens was not seen, compared to the repeated water-soak test where many samples developed cracks during testing. As with the repeated water-soak test, the first soaking cycle may give different results to subsequent soaking cycles on the same sample, so repeated cycles of soaking and equilibrating samples could be trialed. One likely reason for the difference in swelling behavior between the first, and subsequent soaking cycles is the relaxation of any stresses in the wood, and the effect of any differences in drying history (e.g., sorption hysteresis), which is no longer present after the first soaking cycle. For modified wood, there is the additional possibility of compounds leaching from the sample during the first water soaking, and this affecting the stability in subsequent soaking cycles. This leaching could be from unreacted modification chemicals, or thermal breakdown products of the wood cell wall, as seen in thermal modification [27].

The swellometer test quantifies the rate at which samples swell, as well as the overall level of swelling. It was able to distinguish differences in swelling rate and overall levels of swelling between the different modifications tested here, and the unmodified controls. This provides additional information over the repeated water-soak test, and depending on the application, it may be appropriate to perform both the repeated water-soak and swellometer tests, or just the swellometer test.

### 4.3. Equilibrium Humidity Cycling

For both the control and AC samples, results from the equilibrium humidity cycling test were similar to those from the repeated water-soak test. However, in comparison with the ASE results (Figure 2), this test showed clearer differences between the behavior of the TH and FA samples, with the TH samples swelling significantly less than the FA samples. Swelling in humid air occurs at lower moisture contents than soaking in water, so it is plausible that swelling behavior in humid air could be different to liquid water, and that this behavior could differ between modifications or between tree species. The control samples swelled by around 0.045% in the tangential direction for every 1% increase in relative humidity, compared to a less than 0.01% change for the AC samples. Changes in dimension for each 1% change in relative humidity is fairly straightforward to relate to in-service behavior, and this test can easily be applied to modified wood as well as unmodified wood from different species. EMC was also significantly reduced in all modifications, compared to the control samples. Additionally, the AC samples had significantly lower EMC values compared to the other modifications.

Compared to the repeated water-soak test, the calculation of anisotropy ratios was much easier due to having multiple measurement points per specimen, which minimized the overall effect of individual measurement errors and enabled anisotropy ratios to be

calculated for almost all specimens. This included the AC modification where the low levels of swelling made it impossible to calculate meaningful ratios from water-soaking tests. These results did not show any significant differences in swelling anisotropy ratio between the different wood types. The equilibrium humidity cycling test is relatively time consuming to perform—with long equilibration times required at each humidity condition, and each humidity condition needing to be maintained for long periods. This either requires multiple humidity chambers at different relative humidity conditions or requires exclusive use of an adjustable humidity cabinet for long periods (often a year or more). For these reasons, equilibrium humidity cycling is not particularly suitable for frequent screening tests, as is required during the development phase of new wood modifications. However, this test gave useful information on relative levels of swelling with changes in air humidity, as well as equilibrium moisture contents over a range of relative humidity levels and enabled the calculation of anisotropy ratios in samples that had very high dimensional stability, such as Accoya.

### 4.4. Harris Humidity Cycling

This test showed significant differences in the level of swelling between the control samples and the different modifications, with the average swelling of the control samples being over three times higher than that of the modified samples. There were no significant differences in the level of swelling between the three modifications, which contrasts with the equilibrium humidity cycling test, which showed significant differences between each of the modifications. This suggests there may be differences in swelling behavior over long and short time periods, similar to what was seen in the two water-soaking tests. Random variation from measurement errors appears to dominate some of the measurements in this test, due to the smaller dimensional changes being measured. This resulted in some specimens showing negative dimensional changes (i.e., shrinkage) after exposure to the high humidity conditions. This variability makes it more difficult to determine differences between wood types, and to understand underlying behaviors. It is also possible that the paint used to prevent moisture movement through the wood faces is having some restraining effect on the wood swelling. This could be tested by allowing the samples to come to equilibrium at high humidity and comparing their dimensions to the equilibrium humidity cycling specimens. Refinements to this test could reduce the level of scatter in the results (e.g., larger sample dimensions and longer time at 90% RH) and give a clearer picture about the short-term swelling behavior in humid air. The Harris test takes less time to perform than the equilibrium humidity cycling test, but preparing the samples is much more labor intensive, requiring the surfaces to be sealed, and the testing still takes several months, and requires exposure to several humidity conditions during this time. Given the differences in swelling rate during water soaking, it is likely that there are similar differences in swelling rates in humid air. Measuring dimensions over time during exposure to humid air would provide interesting insights into these swelling rates and would likely show if the painted surfaces were preventing the wood from swelling. The combination of long time periods to equilibrate at different humidity levels, and the requirement for specialist measurement equipment limiting the number of samples that could be tested at once would make this a very time-consuming test to perform.

### 5. Conclusions

For all the tests compared here the modified samples had significantly greater dimensional stability than the unmodified control samples, but depending on the test, the three modifications performed differently relative to each other, suggesting that dimensional stability behavior can vary depending on the conditions the samples are exposed to, and the time of exposure to these conditions.

In the repeated water-soak test the FA and TH samples did not have significantly different levels of swelling, but the swellometer test showed substantial differences in the rate of swelling between the two modifications. The maximum swelling measured in the

swellometer test correlated well with the tangential swelling in the repeated water-soak test, suggesting that the swellometer test could be used as a standalone test to determine both the rate, and the overall degree of swelling in a single test.

In the equilibrium humidity cycling test, the different modifications performed similarly, relative to each other, to the repeated water-soak test, but differences between the TH and FA samples were significant in the equilibrium humidity cycling test, where they were not significant in the repeated water-soak test. The equilibrium humidity cycling test enabled the calculation of swelling anisotropy ratios in samples with high dimensional stability, which was not possible from the repeated water-soak test data.

The Harris test did not show any significant differences in swelling between the different modifications, which is in contrast to the results from the other tests, almost all of which showed significant differences between the modifications. As with the swellometer test, measuring the dimensions continuously during exposure to humid air would provide additional insights into the rate of swelling of the different modifications.

All the tests compared here provided unique information on the dimensional stability of the different wood modifications. Tests that capture the rate of dimensional change provide an additional insight into the wood behavior, compared to tests that only measure equilibrium conditions. There may be differences in swelling behavior in liquid water compared to that in humid air, and for some applications (e.g., indoor use) the behavior in humid air is of primary interest, so it is important to be able to quantify this behavior. As with many types of property testing, choosing an appropriate dimensional stability test is a trade-off between managing the duration and complexity of the testing, and the need to measure conditions similar to those the wood will see in service, and it is unlikely that one single test will meet all these requirements.

**Supplementary Materials:** The following supporting information can be downloaded at: https://www.mdpi.com/article/10.3390/f13040613/s1, A document with additional details of the data analysis is published online. This includes the following sections: Section S1: Linear regression to predict anti-shrink efficiency (ASE) from tangential swelling; Section S2: Examples of swellometer swelling rates over time; Section S3: Comparing maximum tangential swelling from water-soak and swellometer tests; Section S4: Calculation of swelling coefficients from equilibrium humidity cycling test and Section S5: Calculation of swelling anisotropy ratios from equilibrium humidity cycling test.

**Funding:** This research was funded by The Ministry of Business, Employment and Innovation New Zealand, through the Strategic Science Investment Fund (Grant number C04X1703). The APC was funded by the same fund.

**Institutional Review Board Statement:** Not applicable.

**Informed Consent Statement:** Not applicable.

**Data Availability Statement:** The data presented in this study are openly available in FigShare at 10.6084/m9.figshare.19333583.

**Acknowledgments:** The following people assisted with sample preparation and experimental work: Maxine Smith, John Turner, Hank Kroese, Tatjana Smolic, Meeta Patel, and Sheree Anderson. Warren Grigsby developed the furfuryl alcohol formulation used to modify the FA samples. Gavin Durbin developed the Swellometer method. Elizabeth Dunningham, Ayyoob Arpanaei and Steve Riley assisted with the preparation of the manuscript. The contributions of all these people are greatly appreciated. The ThermoWood radiata pine was purchased from Tunnicliffe's Timber in Edgecumbe New Zealand and the Accoya radiata pine was purchased from ITI Timspec New Zealand.

**Conflicts of Interest:** The author declares no conflict of interest. The funders had no role in the design of the study; in the collection, analyses, or interpretation of data; in the writing of the manuscript, or in the decision to publish the results.

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
