# Peer review of "Evaluating Dimensional Stability in Modified Wood: An Experimental Comparison of Test Methods"

_forests, doi:10.3390/f13040613_

Round 1

Reviewer 1 Report

Thank you for submitting this clear and interesting study as a paper. Comparisons of this type are potentially very useful in highlighting differences between measurement techniques and modification systems.

In one case you have taken a well established technique and possibly obtained unusual results through adapting the method, and in another, you have taken a less commonly used method and made potentially great progress by extending the monitoring period to capture new and interesting data. In both cases I think there is scope to dig a little deeper.

For the water soak (ASE) test, you seem to have used 'biscuits' of wood, no dimensions are given, but can I assume that these were slices cut from the plank with approx cross section of the plank (if the second swelling method info is an indicator, perhaps this was 38x100mm, or perhaps based on the plank data  it could have been 100x50mm, but it would be useful to know [please add this to the revision]). This is unusual in an ASE test - most of the recent literature has settled on smaller dimensions, e.g. 20 x 20 x 5mm in ref 9, but sometimes larger, e.g. 30 x 30mm and possibly up to 10mm longitudinally (although the longer this dimension becomes the less relibale the test will be). It seems the biscuit thickness in longitudinal direction was 15mm - this is quite high for an ASE test. Did you consider that by using biscuits rather than smaller pieces you would be introducing error from growth stresses and 'fixed' kiln dimensions in the wood? and that potentially the large dimensions would start to introduce other restrictions on the pure tangential and pure radial swelling case? For small samples there is less hindrance due to size and shape, or cupping or other plank level effects seen when wetting large pieces.

With that comment made, I do wonder whether some of the odd effects seen for T/R ratio in acetylated Radiata was due to the size of the test samples, rather than a what would have been seen for smaller cubes. Pervious studies in other labs have never flagged that radial swelling in acetylated wood drops so significantly.

On the subject of method data, I understand that if commercial processes are being used, it may not be possible to give full details about the treatment, however the Radiata which was furfurylated at Scion perhaps you have more information than this available? Or a reference which could be given to refer the reader to standard details for this process (as opposed to using Kebony or other commercial product)?

On the water soaking process for the ASE test, it also appears that you have submerged in water then used vacuum, possibly releasing then applying pressure. This is a fairly low efficiency method for such thick samples. A vacuum prior to immersion would be more likely to succeed perhaps. Or a vacuum and pressure as would be done in pressure treatment - but this is rarely needed for ASE tests.

It is also intersting that you chose to not use the 'control' samples within the calculation of ASE values for the modified woods. Instead you used literature data. If this is the case, it could have been equally viable to stop at the swelling coefficient stage (e.g. in ref 9, S% is calculated for the mod and the unmod samples, and both values are used to determine ASE for the mod wood) rather than go and do an ASE with textbook data. I think the approach is interesting, but didn't see enough discussion of why this approach had been used, and the other had been rejected. e.g. if the same wood was used for the furfurylation as the control, in this case you could have both a traditional ASE value as well as the 'literature fudge' ASE value available, and could make a comparison, even if the commercial TH and AC material was not matched.

Short term water soak (swellometer)  - here the frequent measurements and different intervals for calculating ASE and swelling in T and R direction holds great potential, and the sample is a more sensible 6mm thick to allow water ingress to be relatively unhindered. But again you have opted to use literature data rather than the control boards, and as a result have achieved negative ASE for the controls - but not discussed this much. Essentially it could appear that between batches of unmodified Radiata, a variation of up to +/-20% ASE could be discovered, simply by having unmatched controls. This is quite a big issue! I know from practical experience this to be true. Hence, determining a method that is robust even when working with commercial or unmatchable material is quite important. But is using a literature value a valid solution? Based on this result I find it questionable. I think there is plenty to discuss here.

Equilibrium humidity cycling test - these samples are a much better size than the 'biscuits' in ASE tests. Method clarification, on line 214, possibly an extra few words are needed to say that all steps in Table 1 were completed as one cycle, then the full table was repeated for subsequent cycles?

For the calculation of h (swelling coefficient from eq humidity experiment), did you account for differences between the ranges, e.g. from 60 and 90% and between 60 and 30% - some studies have indicated that rate may be different at earlier parts of the swelling-m.c. curve than at later parts of the curve? Possibly this is for a later paper, not to add here, but could be acknowledged if you have considered it. Also, did you consider hysteresis? was the rate on sorption different than on desorption? or was it only calculated on one direction not the other when doing the cycling? If so perhaps indicate this.

In the Harris test. Are you confident that the epoxy sealant is sufficiently able to stretch with the swelling of the wood? or able to shrink without hindering the shrinkage of the wood? Also, are you confident that the dimensions of these samples are going to allow water vapour to reach an equilibrium in a suitable time frame? Access is from the edges, and the dimension of 30mm means a total distance of 15mm must be travelled to reach the core. the equilibrium would be reached faster and/or lower likelihood of prematurely truncating the measurements if dimensions were smaller. In a 24 hour window of time I wonder if the modified woods have had enough chance to start taking up moisture? given that in full imersion you have observed significant differences in uptake rates - at 30mins several were a long way from their fully swollen state. In air with humidity this is likely to take longer. (more work for a subsequent study).

Based on the use of biscuits not small square (truly aligned Tang x Rad) samples, and a submerssion prior to vacuum, plus later mention of examples where untreated material is not available, I can only assume that you were attempting to find a process that is suited to commercial production, or an industrial context, but this narrative wasn't well drawn out in the introduction or discussion. In such cases, an indicative test might well be justified, and if so a comparison with a more 'pure' lab-style ASE test would be a very interesting additional study.

In lines 317-324 you start to hint at this being case. However, wouldn't S% just do this job? and avoid putting a source of error (literature value) in place of the matched controls to try and calculate ASE. Surely a literature value is as bad and/or as potentially good as a unmatched control of the same species? Just as likely to be off the mark. Perhaps this needs to be discussed (e.g. on page 13). ASE is ultimately a metric used by wood scientists to check their clever modification systems while doing lab work. Swelling coefficients are well established for use with joiners and carpenters and practical wood applications. See Wood Handbook from USDA or Otto Suchsland Swelling and Shrinking of Wood.

In line 396-398 you comment about the different EMC values for the modified woods, it is true that all would struggle to reach 12% m.c., and this is well known. But you haven't mentionned that many studies actually use the standard conditions as the target (i.e. 20C and 65% r.h.) not the 12% m.c. which arises. So for modified woods using 20C and 65% r.h. would be considered equivalent, even though the EMC at those standard conditions would be lower (as shown in your Fig 5). There is a further complication that some modifications contain other composunds, e.g. the FA modification, which may be considered as a separate phase, adding weight to the total wood weight and artificially lowering the m.c. value. This is more pronounced for polymers e.g. PMMA, or oils, but may occur with e.g. FA or formaldehyde resin systems too.

I hope that the above points help in further refining this paper. It is a good piece of work

Author Response

I have added the dimensions of the ASE biscuits and some discussion of the advantages and disadvantages of different specimen sizes.

I assume the comment on T/R ratios for Accoya not matching published figures is referring to the ratios calculated from the water saturated dimensions.  For the T/R ratios calculated for the water soaking, the dimensional changes measured in the Accoya samples were incredibly small, and had very high levels of experimental errors (e.g. many samples showing negative dimensional changes when soaked in water) and it was concluded that these errors were “making the ratio meaningless for this wood type”. This being the case, I don’t think it is necessary (or helpful) to compare these values to published figures because they are not measuring something meaningful. On the subject of smaller samples potentially giving better quality results; using smaller samples with the same dimensions in both the radial and tangential directions would mean that the experimental errors were similar in the radial and tangential direction, but it would not change the fundamental issue that the dimensional changes being measured were too small for the measurement technique being used (indeed it would increase the measurement errors in the tangential direction, due to the decreased tangential dimensions). For the T/R ratios calculated from the humidity cycling test, I have included a reference that also measured reduced shrinkage anisotropy in acetylated samples. This suggests that, as far as the AC samples are concerned, the ratios calculated from the humidity cycling are much more believable than those calculated from the water soaking test.

I have added some details on the FA process (composition of the formulation, and impregnation process used).

I’m a bit confused about the comments on the resaturation method we used, and the suggestion that a pressure treatment may have been better. The vacuum/pressure resaturation method we use is almost identical to a ‘tank’ treatment used to impregnate small volumes of wood with preservatives or modification agents – the wood is placed in a vessel filled with the impregnation solution (water in this case) it is placed in a pressure vessel, vacuum is applied for a certain period of time, and then pressure is applied to the vessel for a period of time. As well as being a common method of impregnating wood, this appears to be used in a number of other studies measuring dimensional stability (e.g. García-Iruela, A., Esteban, L.G., García Fernández, F., de Palacios, P., Rodriguez-Navarro, A.B., Martín-Sampedro, R., Eugenio, M.E. (2019) Effect of vacuum/pressure cycles on cell wall composition and structure of poplar wood. Cellulose. 26:8543-8556.) Following this vacuum/pressure resaturation plus water soaking the wood samples all sink, suggesting they are well saturated with water.

On the use of published literature values as an ‘unmodified reference’ for the ASE and swellometer calculations – I am aware that this is not typical, and I am not suggesting that this is the best way to analyse the data presented here. As mentioned in the paper, ASE and ASEw are very good for comparing the behaviour of modified wood relative to a matched unmodified control. The data presented here is illustrating the types of results you obtain from these calculations, with the complication that there is no obvious way to select an ‘unmodifed reference’ for the set of data here. It could just as easily have been calculated from the average values of the unmodified control samples, which would mean that some samples were matched with the controls and some were not. Instead I decided to use a published figure, so all the treatments (including controls) were being compared on a consistent basis. I have clarified the methods and results to explain that the ASE is the preferred calculation where unmodified reference measurements are available, otherwise a different calculation should be used. I have done the same for the swellometer test. Overall the choice of a different reference (Control samples vs published figure) will just shift each of the ASE or ASEw values on a proportional basis, it won’t actually change the relationship between the different modifications.

I have added some extra details to clarify the sequence of humidity conditions used in the equilibrium humidity cycling (listed in Table 1).

For the calculation of h – the coefficient is a line fitted through both the adsorption and desorption data, so is an average of the shrinking and swelling. I have added a sentence to the methods to clarify this. Swelling and shrinking rates do vary with moisture content, but the difference in slope from 90-65%RH and 65-30%RH was relatively small, so a single slope has been calculated over the entire RH range. Above 90% RH and below 30% RH the sorption isotherms do generally change slope considerably (especially at very high RH), but this is outside the range being measured here, and outside the range of humidity conditions that would typically be seen in service.

On the Harris test, I have included some discussion on the possible restraining effect of the expoxy sealer, and a way you could assess if that was occurring. In terms of whether the surface area and time of exposure would be sufficient for the samples to come to equilibrium – they almost certainly would not have come to equilibrium. The intention of this test is to assess behaviour after short periods of exposure to very high or low humidity conditions, not to understand equilibrium conditions (which are covered by the equilibrium humidity test). I have added a sentence to the methods section to clarify this. Ideally this test would be performed similarly to the Swellometer test where dimensions are monitored over time, rather than being measured at a single point in time.

I have added some details to the methods about the choice of swelling coefficient (%S) vs ASE vs tangential swelling (%T) as the preferred metric for understanding dimensional changes between water saturated and oven dry conditions. While %S is widely reported (ether from green or water saturated conditions to oven dry conditons, or to 12% MC), I do not believe this gives an accurate picture of how the wood will perform in service, where moisture changes are likely to be over a smaller moisture content range, and the degree of movement in particular grain directions is likely to be more important than the overall change in volume of the wood. In this paper I have aimed to present a range of test methods and calculations, some of which will be suited to wood modification research, and some will be suited to understanding the in-service behaviour of different types of wood. That said, all are intended for use in a research context, not as a rapid industrial style test, as has been suggested by the reviewer. Unfortunately, from my experience, the more applicable a test is to real world wood behaviour, the more time consuming the test method (e.g. equilibrium humidity cycling).

I have added some comments about test methods requiring samples to be equilibrated at standard conditions, irrespective of the actual moisture content achieved at those conditions, and I have included a comment about the AC and FA samples having a low apparent moisture content due to the increased mass of the wood plus modification agents.

Reviewer 2 Report

The author made an interesting approach. He tried to evaluate the dimensional stability of thrtee different types of modified wood. The methodology is sound and the discussion of the results is adequate. Interesting conclusions have been come out from this paper. The main drawback of the ppaer is the poor literature review in the properties (dimensional stability) of the modified wood. This has to be extended

Author Response

I have extended the introduction to give background information on the mechanisms of dimensional stability for each of the three different wood modifications and described two studies which compared dimensional stability results between these modifications. This gives an idea of the differences in behaviour that may be seen between the different modifications. The aim of this paper is to get a better understanding of the dimensional stability test methods, using the different wood modifications as a standardised way to illustrate these results. Understanding how the different modifications behave relative to each other, and relative to the controls is more important than the individual dimensional stability results obtained. With this in mind, I haven’t provided detail on levels of dimensional stability that have previously been found for each modification type, because the intention of this paper is to compare properties between modifications and between tests, not to quantify the dimensional stability of each modification.